# Mesoporous Organosilica Nanoparticles to Fight Intracellular Staphylococcal Aureus Infections in Macrophages

**DOI:** 10.3390/pharmaceutics15041037

**Published:** 2023-03-23

**Authors:** Manasi Jambhrunkar, Sajedeh Maghrebi, Divya Doddakyathanahalli, Anthony Wignall, Clive A. Prestidge, Kristen E. Bremmell

**Affiliations:** Centre for Pharmaceutical Innovation, UniSA Clinical and Health Sciences, University of South Australia, Adelaide, SA 5000, Australia

**Keywords:** antibacterial, mesoporous silica nanoparticles, rifampicin, intracellular infection

## Abstract

Intracellular bacteria are inaccessible and highly tolerant to antibiotics, hence are a major contributor to the global challenge of antibiotic resistance and recalcitrant clinical infections. This, in tandem with stagnant antibacterial discovery, highlights an unmet need for new delivery technologies to treat intracellular infections more effectively. Here, we compare the uptake, delivery, and efficacy of rifampicin (Rif)-loaded mesoporous silica nanoparticles (MSN) and organo-modified (ethylene-bridged) MSN (MON) as an antibiotic treatment against small colony variants (SCV) *Staphylococcus aureus* (*SA*) in murine macrophages (RAW 264.7). Macrophage uptake of MON was five-fold that of equivalent sized MSN and without significant cytotoxicity on human embryonic kidney cells (HEK 293T) or RAW 264.7 cells. MON also facilitated increased Rif loading with sustained release, and seven-fold increased Rif delivery to infected macrophages. The combined effects of increased uptake and intracellular delivery of Rif by MON reduced the colony forming units of intracellular SCV-SA 28 times and 65 times compared to MSN-Rif and non-encapsulated Rif, respectively (at a dose of 5 µg/mL). Conclusively, the organic framework of MON offers significant advantages and opportunities over MSN for the treatment of intracellular infections.

## 1. Introduction

Intracellular infections, where bacteria enter and colonize host cells, result in recurring infections such as tuberculosis and urinary tract infections. Intracellular bacteria and fungi can elude their host’s immunity and effectively survive against conventional antimicrobial treatments due to (1) the inability of antibiotics to penetrate cell membranes [1], (2) their localization in specialized compartments of host cells (i.e., phagosomes, lysosomes or the cytosol) with specific localized environments (e.g., pH, enzymes and nutrients) which renders antibiotics ineffective [2], and (3) cellular efflux of antibiotics [2,3,4]. Intracellular infections are commonly recurrent which can pose a significant challenge of increasing the minimum inhibitory concentration (MIC) of antibiotics resulting in antibiotic resistance [3]. Small colony variant (SCV) *Staphylococcus aureus* (*SA*) bacteria are one of the many intracellular pathogens that result in a plethora of life-threatening conditions such as endocarditis, pneumonia, and sepsis [5]. SCV *SA* differs from wild-type *SA* by virtue of (a) slow growth [6], (b) small colonies on agar plates [7], (c) the colonies show reduced or absence of pigmentation [7], (d) modified pattern of virulence factors (e.g., reduced production of a-hemolysin [α-toxin]) [6], and (e) altered drug resistance profiles resulting in challenges for their effective treatment [5,8]. Thus, they prove a challenge for designing an effective approach to treat intracellular infections.

Various antimicrobial nanocarriers have been reported to be efficacious in treating intracellular infections by virtue of their efficient intracellular delivery of cell impermeable antibiotics [9,10,11]. Their potential for cell internalization through the phagocytic/endocytic pathway assists in the release of antibiotics at the location of infections, i.e., phagosomes [12]. The effective delivery vehicle for intracellular infection is dictated by the composition of the nanocarrier, i.e., size, surface charge, and surface functionalization [13]. Mesoporous silica nanoparticles (MSN) are highly porous materials with application in many fields such as separations, catalysis, biotechnology, and drug delivery as recently considered in numerous review articles [14,15]. MSN have been applied to drug delivery due to their highly porous structures enabling drug encapsulation and stabilization [10,16,17,18]. Post-modification has been explored to alter surface characteristics, i.e., charge and hydrophobicity to improve drug loading, release, and interaction with biological cells [12,18,19]. Recently, an organo-modified meso-porous silica nanoparticle (MON) with organic and inorganic hybrid composition has shown promising efficacy in drug delivery and cancer immunotherapy [20]. Compared with the post-modification method of MSN, the MON framework does not block porous channels, thus endowing an obvious advantage over other modification methods of MSN. Further, MONs possess additional advantages, i.e., (1) desirable physiochemical features such as hydrophilicity/hydrophobicity balance [21,22], (2) incorporation of organic groups in the MON framework reduces steric hindrance for encapsulation of the guest molecules [23], and (3) easy biodegradation [24,25].

Herein, we report for the first time the application of MON as an intracellular antibacterial drug delivery platform for the effective treatment of SCV *SA*. More specifically, we investigated the efficacy of ethylene bridge MON on the delivery of the challenging antibiotic rifampicin (Rif) for treatment of SCV *SA* infections in macrophages (schematic in Figure 1). Although Rif has demonstrated efficacy against intracellular SCV *SA* infections, its use has been hindered due to low solubility and permeability [26]. Rif loading and release, cellular interactions and toxicity, and intracellular antibacterial activity of the formulations were determined. By the merits of the anion swelling synthesis protocol [27], the MSN and MON that were synthesized were of similar size, thus enabling comparison of the antibacterial effect of the organosilica framework with the inorganic framework for cellular delivery of Rif to treat intracellular infections.

## 2. Materials and Methods

### 2.1. Chemicals

Cetyltrimethyl bromide (CTAB), tetraethyl orthosilcate (TEOS), 1,2-bis(triethoxysilyl)ethane (BTEE), triethanolamine (TEA), sodium heptaflurobutyrate (FC4), phosphate buffered saline solution (PBS) tablets, and bovine serum albumin (BSA) were purchased from Sigma-Aldrich (Castle Hill, Australia). Ethylene diaminetetraacetic acid (EDTA), hydrochloric acid (HCl), methanol, dimethyl sulfoxide (DMSO), and ethanol were purchased from Ajax Finechem Pty Ltd., Sydney, Australia. Dubelco’s modified Eagles medium (DMEM), 4′,6-diamidino-2-phenylindole (DAPI), penicillin−streptomycin antibiotic mixture, fetal bovine serum (FBS), rhodamine B (RITC), and phosphate- buffered saline (magnesium- and calcium-free) were purchased from Sigma-Aldrich (Castle Hill, Australia). Phalloidin Alexa Fluor 488 dye, tryptone soya agar (TSA), 3-(4,5-dimethylthiazol-2-yl)-2,5-diphenyltetrazolium bromide (MTT), and triton-X 100 were purchased from Thermo Fischer Scientific (Adelaide, Australia). Rifampicin powder was purchased from Sigma-Aldrich (Castle Hill, NSW). RAW 264.7 and HEK 293T cells were obtained from American Type Culture Collection (ATCC, Manassas, VA, USA). SCV *SA* strain WCH SK2 was provided by Dr. Stephen Kidd (University of Adelaide, Adelaide, Australia) [28]. 

### 2.2. Synthesis of Mesoporous Silica Nanoparticles (MSN)

MSN were synthesized based on a previous literature method with small modifications [29]. Briefly, 0.068 g of TEA was stirred with 25 mL distilled water in an oil bath for 30 min. To the resultant mixture, 380 mg of CTAB and 125.3 mg of FC4 was added and stirred at 80 °C for 1 h. Further to the resultant mixture, TEOS (4 mL) was added and the reaction continued for 2 h. The products were collected by high-speed centrifugation (20,000 rpm for 20 min) and washed several times with ethanol. Finally, the surfactant was removed by extraction in concentrated sulfuric acid:methanol (1:7) where it was left to reflux overnight at 70 °C. The suspension was then centrifuged, washed, and dried in the oven.

### 2.3. Synthesis of Mesoporous Organosilica Nanoparticles (MON)

MON was synthesized based on a previous report with slight modifications [30]. Briefly, TEA (0.068 g) and distilled water (25 mL) was stirred at 80 °C in an oil bath for 0.5 h. After 0.5 h, 380 mg of CTAB and 90 mg of FC4 was added and the reaction continued. After 1 h of stirring, 2 mL of TEOS was added with 1.8 mL of the organosilica precursor BTEE. The reaction was continued for 24 h and the reaction products were collected by centrifugation at 20,000 rpm for 20 min. The reaction products were washed, and the material was dried at 50 °C. The resultant particles were extracted as described for MSN.

### 2.4. Structural Characterisation of MSN and MON

Structural characteristics of the MSN and MON samples were observed using a field emission scanning electron microscope (SEM, Carl Zeiss Microscopy Merlin with GEMINI II, Oberkochen, Germany) operated at 2.5 kV and a transmission electron microscope (TEM, JEOL, JEM-2100F-HR) operated at 200 kV. Nitrogen adsorption-desorption isotherms were determined using Micromeretics Tristar II 3020 Surface Area and Porosity Analyser. Particle size distribution and z-average diameter were determined by dynamic light scattering (Zetasizer Nano, Malvern Instruments, Worcestershire, UK) by diluting the particles in PBS at 37 °C. 

Nitrogen adsorption/desorption isotherms were collected using a Micromeritics TriStar II volumetric adsorption analyzer (Micromeritics Instrument Corporation, Norcross, GA, USA) at liquid nitrogen temperature. The particle sample was first outgassed for 3 h at 200 °C. Brunauer–Emmett–Teller (BET) was used to calculate the surface area from the isotherm in the relative pressure (P/P_o_) range of 0.05 to 0.3. 

### 2.5. Rifampicin Loading and Release Determination

For the Rif loading step, 1 mg/mL of Rif was added to 1 mg of MSN and MON in 1 mL of ethanol. The mixture was placed on shaker at 200 rpm for 6 h. Following which, the supernatant was collected after centrifugation at 15,000 rpm for 10 min and analyzed for Rif using ultraviolet-visible spectrophotometer (UV-VIS) at a wavelength of 254 nm (Thermo Fischer, Waltham, MA, USA).

In vitro release was determined in PBS at 37 °C. Briefly, MSN-Rif and MON-Rif were suspended in 10 mL PBS (sink conditions) and agitated. The samples were withdrawn and centrifuged at 38,000 rcf for 10 min. The supernatant was quantified for Rif using UV-VIS spectrophotometry as described above.

### 2.6. Rhodamine (RITC) Loading

MSN and MON particles (1 mg/mL) were dispersed in Milli-Q water via sonication. RITC stock solution (1 mg/mL in Milli-Q water) was added at the mass ratio of 1:100 (dye to particles) and stirred for 2 h. The particles were separated from free RITC by centrifugation and three washes with Milli-Q water.

### 2.7. Cellular Uptake of MSN, MON and Rifampicin

MSN and MON uptake studies were undertaken in the RAW 264.7 macrophagic cell line using confocal microscopy and fluorescence activated cell sorting (FACS). A total of 2.5 × 10^5^ cells/mL were allowed to adhere overnight in a six well plate at 37 °C in a 5% CO_2_ incubator. After 24 h, the cells were incubated with RITC-labelled MSN and MON at a concentration of 20 µg/mL for 10 h. After removing the supernatant, the cells were washed two times with sterile PBS and then scraped and suspended in FACS buffer (500 mg BSA, 50 mg EDTA, and 100 mL sterile PBS free from calcium and magnesium salts). RITC uptake in the cells was analyzed using flow cytometry with a BD Accuri C6 (BD Biosciences, Franklin Lakes, NJ, USA). The intracellular trafficking of nanoparticles in the RAW 264.7 was recorded using confocal laser scanning microscopy (LSM 510, Carl Zeiss, Oberkochen, Germany) at 8 h. The 2.5 × 10^5^ cells/mL were allowed to attach on the coverslip in a 6 well plate. After 48 h the cells were incubated with 10 µg/mL RITC-labelled particles for 4 h. The cells were then washed with PBS and fixed with formaldehyde. The cells were stained with DAPI (emission wavelength of 461 nm) for the nucleus and Alexa Fluor 488 (emission at 525 nm wavelength) for the cytoskeleton.

Intracellular Rif uptake studies were accomplished by seeding RAW 264.7 cells in 96-well plates at a density of 5 × 10^4^ cells/well. After incubation for 24 h, each well was treated with a 50 μg/mL Rif dose. Samples were collected at 1, 4, and 8 h and centrifuged (600× *g* for 5 min). The pellet (cells) were subsequently washed three times with PBS to remove any extracellular particles adhering to the RAW 264.7 cells surface. The cells were then lysed with 100 μL of DMSO, and the intracellular Rif evaluated by extracting with methanol prior to analysis with high performance liquid chromatography (HPLC) (UFLC XR, Shimadzu, Kyoto, Japan) using a C18 column (Alltech, Lutterworth, UK).

### 2.8. Cell Viability Assays

The toxicity of MSN and MON were tested in vitro using HEK 293T and RAW 264.7 macrophagic cell lines. A total of 5000 cells were seeded in each well of a 96 well plate and incubated at 37 °C and in 5% CO_2_. After 24 h of incubation, the medium was replaced with various concentrations of MSN or MON (200 µL) for 24 h. The medium was then replaced with 100 µL containing MTT reagent for 4 h at 37 °C. MTT is a colorimetric assay, where viable cells reduce the yellow MTT reagent to form purple formazan crystals. The formazan crystals were then dissolved in 100 μL of DMSO. The optical density (OD) was recorded at 570 nm using Tecan, infinite M 200 Pro plate reader and the percentage of residual cell viability was determined.

### 2.9. Intracellular Antibacterial Activity

An intracellular infection assay was achieved according to the method of Clemens et al. [12], with slight modifications. Briefly, RAW 264.7 cells (1 × 10^5^ cells/mL) were seeded in a 24 well plate and incubated for 24 h (37 °C, 5% CO_2_). SCV *SA* bacterial suspension was diluted to a multiplicity of infection of 10:1 from the overnight culture. Following centrifugation (600× *g* for 10 min) of the bacterial suspension, the supernatant was discarded, and the pellet redispersed in DMEM. After 1 h incubation, the cells were rinsed three times to remove any extracellular bacteria. Infected cells were then incubated for 4 h with 1 mL of fresh serum-free DMEM containing either Rif, MSN-Rif, or MON-Rif at a Rif concentration of either 2.5, 5 10, or 12 μg/mL. The media were removed and the cells lysed with 0.1% Triton-X 100 in PBS. The cells were transferred to Eppendorf tubes, serially diluted, and plated on sterile TSA followed by 15 h of incubation at 37 °C. After 15 h, the colony forming units (CFU)/mL were determined.

### 2.10. Statistical Analysis

The experimental data were analyzed statistically using a Student’s t-test (unpaired). Values are reported as the mean ± standard deviation, and the data were considered statistically significant when *p* < 0.05.

## 3. Results and Discussion

### 3.1. Characterisation of MSN and MON

Images of MSN and MON acquired using SEM and TEM, displayed in Figure 2, demonstrate both particles to be spherical with a characteristic porous structure (Figure 2C,D). The results of particle characterization using DLS, TEM, and BET are summarized in Table 1. Hydrodynamic particle diameters of MSN and MON were determined as 115 ± 6 nm and 106 ± 12 nm, respectively, each with polydispersibility indices (PDI’s) of 0.08, indicating uniformly dispersed particles. Corresponding mean diameters of MSN and MON from TEM were 105 ± 10 nm and 96 ± 8 nm, respectively (Table 1), correlating with the DLS results. The surface area provides information on characteristics that are important in describing drug loading and release. MSN had a lower surface area of 442.3 m^2^/g compared to MON with 720.1 m^2^/g as determined using BET. Further analysis of pore size and volume was not undertaken as particle aggregation in the dry state leads to cavities that may contribute to an inaccurate pore size [10]. Larger non-surfactant templated pores may also lead to overestimation of the pore size [31]. An earlier report by Jambhrunkar et al. [30] confirmed that MON with ethylene bridges has a higher carbon (C) content (7.84%) and greater hydrophobicity compared with MSN. Thus, MSN and MON particles were determined to be similar in diameter, while MON was characterized by a higher surface area and pore volume than MSN, suitable for increased drug loading.

### 3.2. Rifampicin Loading and Release from MSN and MON

MON exhibited a Rif loading capacity of 600 µg/mg compared to 135 µg/mg for MSN. The increased Rif loading that was observed for MON correlates with an increased hydrophobic nature and surface area characteristic of MON particles. Higher drug loading provides an advantage for particulate drug delivery, requiring a lower amount of particles to deliver an equivalent dose. In vitro Rif release studies in PBS pH 7 (see Figure 3A) demonstrated faster release from MSN compared to Rif release from MON, i.e., in the first 4 h, 74% Rif released from MSN compared to 20% of Rif released from MON. Rif (Figure 3B) is a lipophilic molecule (Log P 3.7) [32] and has been described as a BCS Class II compound as a result of poor solubility and high permeability [33]. As a lipophilic molecule, Rif would be expected to have a greater affinity for the more hydrophobic surface of MON compared to MSN. The faster, more complete release from MSN may be related to the lower Rif loading level for MSN combined with a more hydrophilic surface facilitating release of the lipophilic Rif molecules.

### 3.3. Cytocompatibility of MSN and MON

Cytotoxicity of the unloaded MSN and MON was tested on murine macrophages (RAW 264.7) and HEK 293T cells using the MTT assay. Murine macrophages were the target cell line for intracellular infection studies whereas HEK 293T was used as a model for normal epithelial cells. Figure 4A,B show that neither MSN nor MON exhibit significant cytotoxicity (cell viability >> 80% at up to 100 μg/mL) with increasing concentration of nanoparticles. This is in agreement with previous studies, where minimal cytotoxicity of porous silica particles has been reported [10,34].

### 3.4. Cellular Uptake

Cellular uptake studies were undertaken with RITC-labelled MON and MSN, using both confocal laser scanning microscopy (Figure 5) and flow cytometry (Figure 6) as tools for visual and quantitative analysis, respectively. In RAW 264.7 cells, there was no significant difference in RITC fluorescence from the untreated and free RITC-treated group indicating minimal cellular uptake of the free dye (Figure 5A,B). However, RITC fluorescence was significantly enhanced in RAW 264.7 cells that were treated with RITC-labelled MSN and MON, as depicted in Figure 5C,D. The highest cellular RITC fluorescence occurred when using MON, indicating the superior cellular uptake of MON compared to MSN. 

A time-dependent RAW 264.7 cellular uptake study was conducted using flow cytometry. Uptake of the RITC-labelled MSN and MON nanoparticles increased from 1 h to 4 h, however after 4 h the increase in particle uptake slowed (Figure 6A). MON (65% RITC positive cells) exhibited 5 times higher uptake in RAW 264.7 cells than MSN (12% RITC positive cells) after 4 h treatment. These results corroborate the understanding that hydrophobicity of the nanoparticles could assist in the cellular uptake. According to Niu et al. [35], hydrophobic nanoparticles exhibit a crucial role in the intracellular fate of the cargo, where silica nanoparticles with a hydrophobic surface modification had higher uptake, 9.0 pg silica per cell compared to unmodified silica nanoparticles of 5.5 pg silica per cell. Thus, hydrophobicity played a vital role in enhancing the intracellular uptake of nanoparticles through hydrophobic interactions with the cell membrane. However, the hydrophobic surface modifications could compromise the cytotoxicity and dispersibility of the silica nanoparticles, thus an alternative surface modification method is needed [35]. In the current study, MON, with intrinsic hydrophobicity, enhanced the uptake of porous silica particles without compromising on toxicity, an obvious advantage over hydrophobic surface modification.

Further, to test the influence of nanoparticles on the intracellular delivery of Rif, RAW 264.7 cells were incubated with Rif, MSN-Rif, or MON-Rif, and the intracellular content of Rif analyzed at definite endpoints (Figure 6B) using HPLC. The intracellular content of Rif increased from 1 h to 4 h, after which it was saturated. At all timepoints, MON delivered the highest Rif concentration to RAW 264.7 cells. At 1 h, Rif had 2% cell internalization compared to MSN-Rif and MON-Rif demonstrating 14.6% and 31% Rif in the cells, respectively. The Rif dose that was determined in RAW 264.7 cells from incubation with MON-Rif for 4 h was 89% compared to 42% from MSN-Rif and 12% for unformulated Rif. In relation to MSN and MON particle uptake studies that were determined by microscopy (Figure 5) and discussed previously, a further increase in the incubation time to 8 h did not have any significant impact on cellular internalization of Rif. Thus, MON, owing to its higher drug loading capacity and hydrophobic properties, improved the intracellular delivery of Rif to RAW 264.7 cells.

### 3.5. Antibacterial Efficacy of the Rif-Loaded MSN and MON against SCV SA 

Ultimately, the ability of the nanoparticles to enhance delivery of Rif to treat intracellular bacteria is of interest. Thus, to test the efficacy of MSN-Rif and MON-Rif on intracellular infections, macrophages were infected with the SCV *SA* pathogen. SCV *SA* is a prototype intracellular infectious pathogen. This pathogen is smaller than the wild-type *SA* and easily infects a macrophage by residing in the endosomal compartment. Maghrebi et al. [9] showed that post-treatment of RAW 264.7 cells with SCV *SA,* the bacteria internalizes. Internalization of SCV SA was confirmed in this study with confocal microscopy (Appendix A), where the RAW 264.7 cell nuclei and SCV *SA* were stained with DAPI (green), and the cytoskeleton were labelled with Alexa Fluor 488 (red). As shown with arrows in Appendix A, the green color indicates the internalized SCV *SA*.

Macrophages that were infected with SCV *SA* were treated with Rif, MSN-Rif, or MON-Rif (Figure 7) as a function of Rif concentration. Rif concentrations between 0 and 12 μg/mL were selected based on the minimum inhibitory concentration (MIC) of 0.125 μg/mL and minimum bactericidal concentration (MBC) of 4 μg/mL against SCV *SA* determined experimentally [10]. Control cells without any treatment did not show any effect on reduction of the bacterial counts. Further, there was no significant difference between the colony-forming units for the control and Rif groups, confirming the poor cellular penetration of unformulated Rif and limited efficacy in treating intracellular infections. In contrast, MSN-Rif and MON-Rif significantly reduced the intracellular bacterial count at concentrations of 2.5 μg/mL and higher. At Rif concentrations of 2.5 μg/mL and higher, MSN-RIF deceased the bacterial colony forming units two-fold compared to unformulated Rif, signifying the importance of a particulate system for the delivery of Rif. This observation agrees with Maghrebi et al. [9] and Subramaniam et al. [10], where particulate encapsulation of Rif significantly enhanced intracellular antibacterial efficacy. A significant finding is that for Rif concentrations of 5 μg/mL, MON-Rif reduced the bacterial colony-forming units in this study by 65-fold as compared to Rif and 28-fold compared to MSN-Rif. Thus, MON with its hydrophobic framework of organic groups enhanced Rif’s ability to treat intracellular infection caused by SCV *SA* compared to MSN. Enhanced antibacterial action was observed from MON-Rif despite the observation of lower in vitro Rif release from MON compared to MSN. Recent studies have noted similar behavior, where MSN enhanced the intracellular antibiotic efficacy, although complete antibiotic release was not observed to occur from the nanoparticle carriers [10,19]. Further in vivo validation would be required, however, MON has demonstrated potential to enable current antibiotics to treat recalcitrant intracellular infections with a reduced dose compared to current clinical practice.

## 4. Conclusions

This is the first report to demonstrate that the organosilica framework of MON facilitates a more effective antimicrobial nanocarrier for rifampicin (Rif) than equivalently sized MSN. In comparison to MSN, MON enabled increased Rif loading and sustained in vitro release, and improved the intracellular delivery of rifampicin to macrophages. The amalgamation of these characteristics facilitated high antibacterial efficacy of MON-Rif against intracellular SCV *SA* in macrophages. This work detailed the relationship between the hydrophobicity of the porous silica particle and antibacterial activity of MON and offers opportunities to improve treatment of recalcitrant clinically relevant intracellular infections. 

The Appendix A is available free of charge at confocal microscopy of intracellular bacteria.

## Figures and Tables

**Figure 1 pharmaceutics-15-01037-f001:**
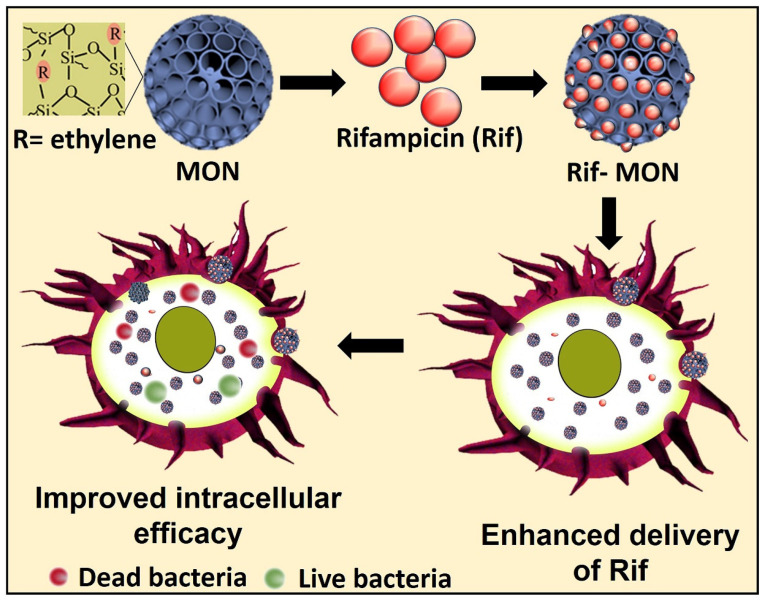
Schematic representation of intracellular antibacterial activity of rifampicin encapsulated in MON.

**Figure 2 pharmaceutics-15-01037-f002:**
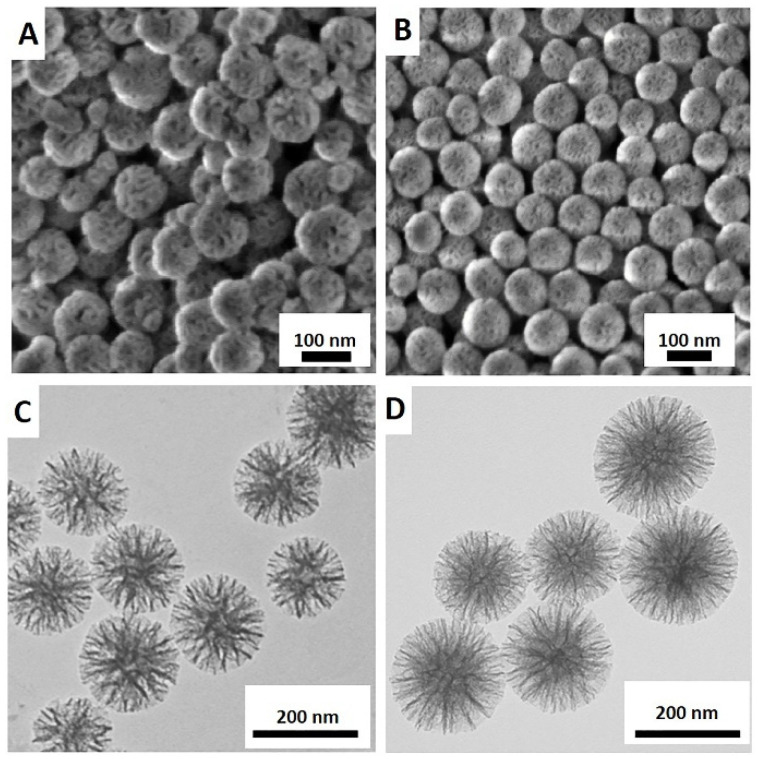
Scanning electron microscopy (SEM) and transmission electron microscopy (TEM) images of MSN (**A**,**C**) and MON (**B**,**D**).

**Figure 3 pharmaceutics-15-01037-f003:**
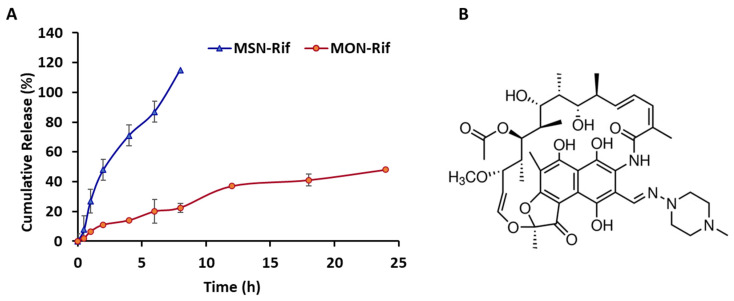
Cumulative release of Rif (**A**) from MSN-Rif and MON-Rif conducted in PBS at 37 °C for 24 h. Data are represented as the mean ± SD (*n* = 3) and rifampicin structure (**B**).

**Figure 4 pharmaceutics-15-01037-f004:**
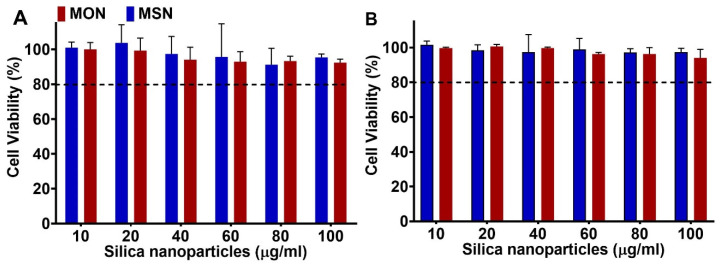
In vitro viability of HEK 293T (**A**) and murine macrophage RAW 264.7 cells (**B**) as a function of MSN and MON concentration. Post-24 h incubation, cell viability was tested using the MTT assay (mean ± SD, *n* = 3).

**Figure 5 pharmaceutics-15-01037-f005:**
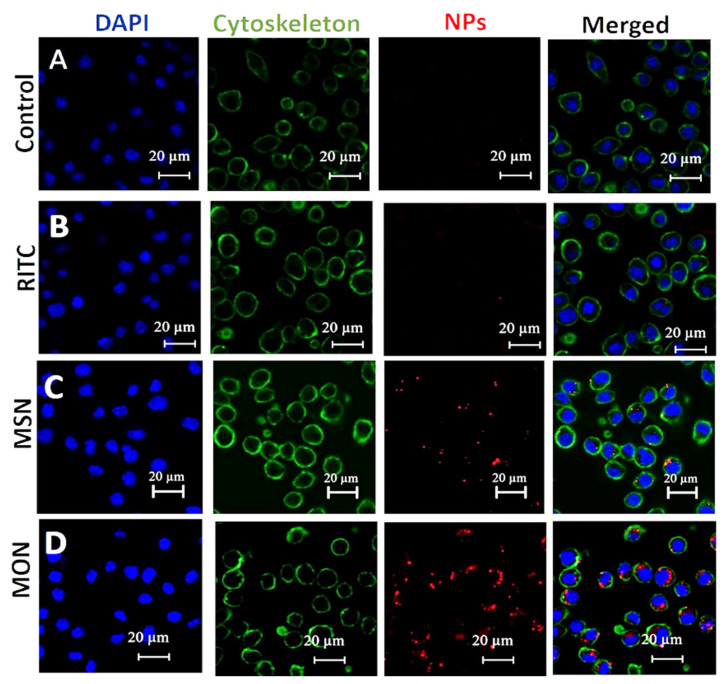
Confocal scanning laser microscopy images of RAW 264.7 cells after 4 h incubation in complete medium. (**A**) Control with no treatment to cells, (**B**) cells that were treated only with RITC, (**C**) cells that were treated with MSN loaded with RITC, and (**D**) cells that were treated with MON loaded with RITC. Nuclei were stained with DAPI (blue), the cellular cytoskeleton was stained with Phalloidin Alexa Fluor 488 dye (green) and the nanoparticles were stained with RITC (red). The control group was treated only with medium whereas all the other groups were treated with the same concentration of the RITC.

**Figure 6 pharmaceutics-15-01037-f006:**
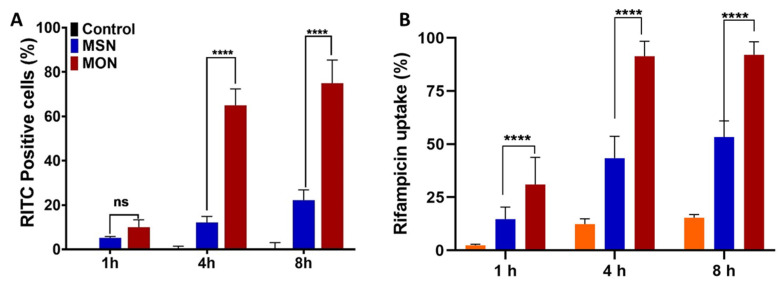
(**A**) Cellular uptake of MSN and MON loaded with RITC in RAW264.7 cells measured with flow cytometry at 37 °C. (**B**) Rifampicin uptake by RAW 264.7 cells; Rifampicin solution (orange bar), MSN-Rif nanoparticles (blue bar) and MON-Rif nanoparticles (red bar), when dosed at a rifampicin concentration of 50 μg/mL (mean ± SD, *n* = 3, ns = not significant; **** *p* < 0.0001).

**Figure 7 pharmaceutics-15-01037-f007:**
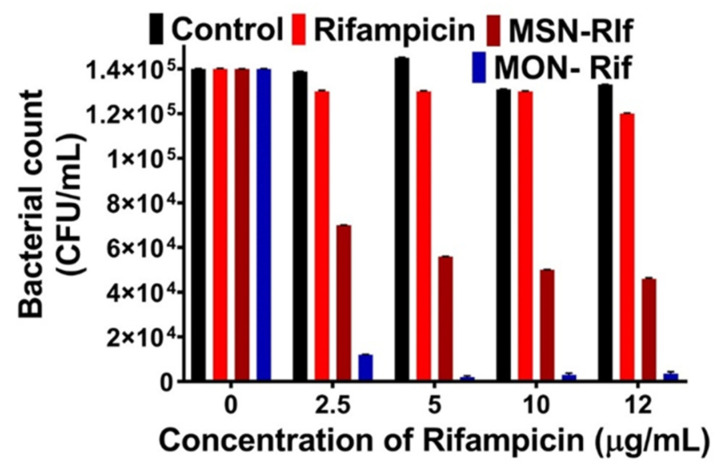
Concentration of the *SCV SA* intracellular viable bacteria after treatment with Rif, MSN-Rif and MON-Rif for 4 h; control cells were untreated.

**Table 1 pharmaceutics-15-01037-t001:** Particle characteristics for MSN and MON.

Particles	DLS (nm)	TEM (nm)	Surface Area (m^2^/g)
MSN	115 ± 6	105 ± 10	442.31
MON	106 ± 12	96 ± 8	720.07

## Data Availability

The datasets generated during and/or analysed during the current study are available from the first author on reasonable request.

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
