# Peer review of "Mesoporous Organosilica Nanoparticles to Fight Intracellular Staphylococcal Aureus Infections in Macrophages"

_pharmaceutics, 2023, doi:10.3390/pharmaceutics15041037_

Round 1

Reviewer 1 Report

The paper can be accepted for the publication in Pharmaceutics after a major revision concerning the following points:

1) The authors should cite the below-reported reviews to provide a more realistic representation of the  possibility of engineering mesoporous silica for nanomedicine and more in general bionanotechnology applications:

Coordination Chemistry Reviews 469 (2022) 214687;

Materials Today Bio 17 (2022) 100472.

2) The structure of Rifampicin should be graphically  represented, if possible highlighting the interactions inside the pores oh the two different mesoporous materials;

3) The discussion on the porosimetric characterization and the presented values related to the pore volumes of the materials are totally wrong and misleading. The pore diameters presented are largely overextimated. The contribution of pore volume of the families of larger, not-surfactant templated mesopores, in both the pore size distributions presented in the supplementary informations, should be excluded from the evaluation of the pore diameter. To the aim of acquiring  some familiarity  with nitrogen adsorption-desorption isotherms the following paper Journal of Porous Materials volume 20, pages 865–873 (2013) can be read and cited.

volume 20pages865–873 (2013)

Author Response

1) The authors should cite the below-reported reviews to provide a more realistic representation of the possibility of engineering mesoporous silica for nanomedicine and more in general bionanotechnology applications:

Coordination Chemistry Reviews 469 (2022) 214687; Materials Today Bio 17 (2022) 100472.

Response: We have included further discussion in the introduction section around mesoporous silica and referenced the two suggested review articles.

Mesoporous silica nanoparticles (MSN) are highly porous materials with application in many fields such as separations, catalysis, biotechnology and drug delivery as recently considered in numerous review articles [14,15].

2) The structure of Rifampicin should be graphically represented, if possible highlighting the interactions inside the pores oh the two different mesoporous materials;

Response: We appreciate this suggestion and have included the structure of rifampicin (in Figure 3B) and improved discussion around the properties of rifampicin and how this would impact on interaction with surfaces of MON and MSN.

“Rif (Figure 3B) is a lipophilic molecule (Log P 3.7) [29] and has been described as a BCS class II compound as a result of poor solubility and high permeability [30]. As a lipophilic molecule, Rif would be expected to have a greater affinity for the more hydrophobic surface of MON, compared to MSN.”

3) The discussion on the porosimetric characterization and the presented values related to the pore volumes of the materials are totally wrong and misleading. The pore diameters presented are largely overextimated. The contribution of pore volume of the families of larger, not-surfactant templated mesopores, in both the pore size distributions presented in the supplementary informations, should be excluded from the evaluation of the pore diameter. To the aim of acquiring  some familiarity  with nitrogen adsorption-desorption isotherms the following paper Journal of Porous Materials volume 20, pages 865–873 (2013) can be read and cited.

Response: We acknowledge the challenges around using nitrogen adsorption data to characterize pore volume and diameter. It has been useful looking into the references the reviewer suggested, in addition to others. According to previous manuscripts, particle aggregation issues may also contribute to errors when undertaking the nitrogen adsorption experiment, in addition to contribution of large non-surfactant templated pores. Thus, as a result we have removed the pore volume and diameter values from Table 1. The pore volume was only referred to once in later discussions on drug loading within the manuscript. The increase in drug loading can be described as due to the increase in surface area and hydrophobicity, and reference to the pore volume has been removed.

Discussion around the pore volume and diameter has been removed from the methods and particle characterization section and the following discussion has been included for the discussion of particle analysis. “Further analysis of pore size and volume was not undertaken as particle aggregation in the dry state leads to cavities that may contribute to the pore size determined [10]. Larger non-surfactant templated pores may also lead to overestimation of the pore size [31].”

Reviewer 2 Report

The authors report on the use of mesoporous silica nanoparticles as drug carriers. For the example chosen (rifampicin against small colony variant staphylococcus aureus, SCV SA) they find that nanoparticles with ethylen bridges (MON) show a much better performance as drug carriers than unmodified silica nanoparticles (MSN). Although my expertise is not in the field of release studies and handling of cells, I find the results convincing and relevant. Overall, the manuscript is written very clearly but the following minor comments should be addressed.

1. I cannot find a definition of the abbreviation PBS.

2. Line 143: "ratio" is presumably "mass ratio".

3. Line 154: What kind of instrument is "BD Accuri C6"?

4. Line 176: I do not understand how dead and live cells are distinguished (sorry for my ignorance).

5. Line 247: Has "flow cytometry" been mentioned in the experimental section? (Perhaps on line 154?)

6. Line 257: "... the particle uptake plateaued." - I don't see a plateau (yet) in Figure 6.A.

7. Figure 6: What is the meaning of the white bars with black frames? What are the meanings of "ns" and "****"?

8. Line 300: I do not understand the sentence "This pathogen is smaller than the SCV." In the previous sentence SCV SA is called the pathogen, SCV was explained as "small colony variants" of it (line 15). So the pathogen is smaller than itself? This doesn't make sense.

9. Figure 7: What do the footnotes in the caption refer to? I see neither "*" nor "***".

10. Conclusions: Is only the difference in hydrophobicity of MSN and MON relevant? Can the authors discuss the relevance of the differences in the structural parameters given in Table 1 (specific surface area, pore diameter, pore volume)?

Author Response

The authors report on the use of mesoporous silica nanoparticles as drug carriers. For the example chosen (rifampicin against small colony variant staphylococcus aureus, SCV SA) they find that nanoparticles with ethylen bridges (MON) show a much better performance as drug carriers than unmodified silica nanoparticles (MSN). Although my expertise is not in the field of release studies and handling of cells, I find the results convincing and relevant. Overall, the manuscript is written very clearly but the following minor comments should be addressed.

  1. I cannot find a definition of the abbreviation PBS.

Response: Thank you for pointing out this omission, we have added the following to the chemicals list in Section 2.1; “phosphate buffered saline solution (PBS) tablets”

  1. Line 143: "ratio" is presumably "mass ratio".

Response: Yes, we have added “mass” to better define this.

  1. Line 154: What kind of instrument is "BD Accuri C6"?

Response: The statement on line 154 has been improved to read; “flow cytometry with a BD Accuri C6 (BD Biosciences, Franklin Lakes, USA).”

  1. Line 176: I do not understand how dead and live cells are distinguished (sorry for my ignorance).

Response: To improve the explanation of the MTT assay, the following sentence has been added to section 2.8. “MTT is a colourimetric assay, where viable cells reduce the yellow MTT reagent to form purple formazan crystals.”

  1. Line 247: Has "flow cytometry" been mentioned in the experimental section? (Perhaps on line 154?)

Response: Thank you for pointing this out, it has now been mentioned according to comment 3.

  1. Line 257: "... the particle uptake plateaued." - I don't see a plateau (yet) in Figure 6.A.

Response: We appreciate that this statement is misleading and have modified the statement to read; “the increase in particle uptake slowed”

  1. Figure 6: What is the meaning of the white bars with black frames? What are the meanings of "ns" and "****"?

Response: The black lines are simply pointing to the data (blue and red bars).

Apologies for the omission, the caption of Figure 6 has been updated to include these definitions. “ ns = not significant; **** p < 0.0001”

  1. Line 300: I do not understand the sentence "This pathogen is smaller than the SCV." In the previous sentence SCV SA is called the pathogen, SCV was explained as "small colony variants" of it (line 15). So the pathogen is smaller than itself? This doesn't make sense.

Response: Thank you for pointing out this typo, it has been amended to “wild-type SA

  1. Figure 7: What do the footnotes in the caption refer to? I see neither "*" nor "***".

Response: You are correct, these were a typo and have been deleted

  1. Conclusions: Is only the difference in hydrophobicity of MSN and MON relevant? Can the authors discuss the relevance of the differences in the structural parameters given in Table 1 (specific surface area, pore diameter, pore volume)?

Response: When considering the significant increase in particle uptake by macrophages observed for MON compared to MSN, we believe that particle hydrophobicity was the major contributing factor, as internal porosity of particles would not impact on particle interactions with a cell membrane. It’s the significant increase in cellular uptake of the MON particles compared to MSN, that results in increased uptake of the encapsulated drug, rifampicin. A much larger study would be required to understand the influence of porosity. We believe the hydrophobicity is the biggest contributor therefore have included this in the conclusion.